# Cautious Gait during Navigational Tasks in People with Hemiparesis: An Observational Study

**DOI:** 10.3390/s24134241

**Published:** 2024-06-29

**Authors:** Albane Le Roy, Fabien Dubois, Nicolas Roche, Helena Brunel, Céline Bonnyaud

**Affiliations:** 1APHP, GHU Paris-Saclay, Raymond Poincaré Hospital, Physical Medicine and Rehabilitation Department, 92380 Garches, France; 2IFMK Saint-Michel, 75015 Paris, France; 3APHP, GHU Paris-Saclay, Raymond Poincaré Hospital, Motion Analysis Laboratory, Functional Explorations Department, 92380 Garches, France; 4Université Paris-Saclay, UVSQ, Research Unit ERPHAN, 78000 Versailles, France; 5Université Paris-Saclay, UVSQ, Inserm Unit 1179, END-ICAP Laboratory, 78000 Versailles, France

**Keywords:** cautious gait, navigation, stroke, stationarity, timed up and go, motion capture, kinematics

## Abstract

Locomotor and balance disorders are major limitations for subjects with hemiparesis. The Timed Up and Go (TUG) test is a complex navigational task involving oriented walking and obstacle circumvention. We hypothesized that subjects with hemiparesis adopt a cautious gait during complex locomotor tasks. The primary aim was to compare spatio-temporal gait parameters, indicators of cautious gait, between the locomotor subtasks of the TUG (Go, Turn, Return) and a Straight-line walk in people with hemiparesis. Our secondary aim was to analyze the relationships between TUG performance and balance measures, compare spatio-temporal gait parameters between fallers and non-fallers, and identify the biomechanical determinants of TUG performance. Biomechanical parameters during the TUG and Straight-line walk were analyzed using a motion capture system. A repeated measures ANOVA and two stepwise ascending multiple regressions (with performance variables and biomechanical variables) were conducted. Gait speed, step length, and % single support phase (SSP) of the 29 participants were reduced during Turn compared to Go and Return and the Straight-line walk, and step width and % double support phase were increased. TUG performance was related to several balance measures. Turn performance (R^2^ = 63%) and Turn trajectory deviation followed by % SSP on the paretic side and the vertical center of mass velocity during Go (R^2^ = 71%) determined TUG performance time. People with hemiparesis adopt a cautious gait during complex navigation at the expense of performance.

## 1. Introduction

Stroke is the leading cause of disability in adults. Locomotor disorders caused by the ensuing hemiparesis significantly impact individuals and their participation in daily activities [1,2,3]. The Action Plan for Stroke in Europe 2018–2030 recommends further research on assessment in people after stroke to develop optimal rehabilitation strategies [4].

The recommended clinical gait assessments after a stroke and for other neurologic conditions are the 10-Meter Walk Test (10-MWT) and the 6-Minute Walk Test (6-MWT). The instrumental gold standard assessment is a three-dimensional gait analysis, widely used to quantify biomechanical gait parameters in people with hemiparesis after a stroke [5,6]. However, all these walking assessments are performed in straight-line conditions, and they do not reflect daily life locomotor tasks, which are goal-oriented, influenced by environmental constraints, and involve changes in direction (accounting for up to 50% of daily displacements) [7,8]. The concept of navigation proposed by Berthoz and Viaud-Delmon better represents daily locomotor tasks [9]. The Timed Up and Go (TUG) test involves navigational tasks: walking to a cone (after rising from a chair), circumventing the cone, and returning to the chair [10]. The TUG test is widely used in people with hemiparesis following stroke and complements straight-line walking performance assessments [10,11].

Stability is defined as the ability of the body to regain a state of balance after a disturbance [12]. The TUG locomotor subtasks, especially the Turn and the preceding phase, challenge the stability of people after a stroke [13,14]. Complex locomotor tasks such as turning are a major cause of falls in people with hemiparesis [15,16,17,18]. As such, some authors consider TUG performance time as an indicator of fall risk in people with hemiparesis, but this is not consensual [14,18,19,20,21,22]. Several biomechanical parameters of locomotor tasks have been found to indicate the risk of falls in people with hemiparesis (center of mass [COM] displacements, percentage duration of single support phase [%SSP] and percentage duration of double support phase [%DSP], and locomotor trajectory) [14,22,23,24]. 

Adaptability during locomotor tasks is crucial for ensuring safety and preventing falls [25]. Previous studies have highlighted gait adaptations in older adults and other people with balance disorders, suggesting the use of a cautious gait strategy to maintain stability [7,26,27,28,29]. The notion of a safer or cautious gait has been proposed in the context of stability control and locomotor adaptations during navigation tasks [29,30]. The concept has been highly studied in older adults and people with Parkinson’s disease and, to a lesser extent, in people after stroke, ataxia, and spinal cord injury [31,32,33,34,35,36]. Studies have shown that people with balance disorders adapt their spatio-temporal gait parameters (STPs) to ensure a stable gait and prevent falls during simple and complex navigational tasks (involving turns or uneven surfaces) [31,36,37,38,39,40]. Older people walking on unstable surfaces and those with balance disorders adopt a cautious gait by decreasing their gait speed, cadence, and/or step length and increasing their step width and %DSP [7,28,29,39,41]. These gait adaptations may be caused by balance disorders or the fear of falling without an actual risk of falling [26,41,42,43]. 

The gait of individuals with chronic hemiparesis is also modified to cope with balance impairments, with asymmetry of temporal and other biomechanical parameters. During straight-line walking, temporal asymmetry is associated with the difficulty of maintaining stability during the SSP on the paretic side [44,45]. People with hemiparesis who fall have a slower gait, slower cadence, and larger SSP percentage asymmetry compared to non-fallers during straight-line walking [23,46]. During a complex walking task involving reaching targets with the knees by performing medio-lateral translations during treadmill walking, participants with post-stroke hemiparesis decreased their gait speed and step length more than able-bodied participants [38]. In addition, during complex turning tasks, people after a stroke (with or without a fall history) also reduced the rotation speed during the Turn and increased the number of steps, as well as deviating their locomotor trajectory compared to age-matched controls [13,14,24,47,48,49]. Similar results were found during preparatory turn and turn tasks: compared to healthy individuals, people with stroke or other brain lesions reduced their gait speed, cadence, step length, and % SSP on the paretic side and increased their step width, number of steps, and % DSP [50,51,52,53,54]. However, STPs during turn and navigational tasks in fallers and non-fallers with hemiparesis are unknown.

In older individuals and people with Parkinson’s disease, these biomechanical changes are considered as a means to maintain stability and compensate for balance disorders [44,48,53,55]. Therefore, biomechanical gait parameters may not only be indicators of fall risk but may also be highly informative of cautious gait. 

Few studies have compared gait adaptations between navigational tasks of varying difficulty in people with hemiparesis [22]. Studies are needed to complete the knowledge of adaptations and adoption of a cautious gait during complex navigational tasks.

The primary aim of this study was to determine if people with hemiparesis adopt a cautious gait during complex navigational tasks by comparing STPs between each navigation subtask of the TUG (oriented gait to the cone [Go], turn around the cone, and oriented gait to the seat [Return]) and a Straight-line walking task in people with stroke-related hemiparesis. We hypothesized that gait speed, step length, and %SSP on the paretic side would be shorter; and step width would be larger and %DSP would be longer in the more complex tasks, indicating a cautious gait. The secondary aim was to better understand this behavior by analyzing the relationships between STPs during the TUG navigational tasks and the clinical measures of balance, confidence, and falls, as well as to compare STP during the TUG navigational tasks and Straight-line walk between fallers and non-fallers. TUG performance time reflects the adaptation strategies of people with hemiparesis. The third aim of this study was to identify the major determinants of TUG performance time among (i) the biomechanical gait parameters during the TUG navigation subtasks (STPs and joint kinematics, COM displacement, and gait trajectory) and (ii) the performances of each navigation subtask of the TUG (Go, Turn, and Return). We hypothesized that (i) the markers of stability and cautious gait and (ii) the Turn subtask would be closely related to timed performance. Our overall aim was to instrument the TUG test with motion analysis to provide comprehensive insights into stroke-related behavioral adaptations during complex locomotor tasks.

## 2. Materials and Methods

### 2.1. Protocol and Registration 

We conducted a prospective, observational study in our movement analysis laboratory at Raymond Poincaré University Hospital (GHU Paris-Saclay, APHP), France.

All study participants provided written informed consent in accordance with the ethical codes of the World Medical Association. The main study was approved by the ethics committee (Comité de protection des personnes Ile de France XI, Ref 13005. CNIL, Ref DR2013-283), and the Research Ethics Committee of the University of Paris-Saclay (CER-Paris-Saclay-2021-086) approved the aims of the present study. The study was registered at the APHP data protection office (registration number: 20231115173304). The study reporting follows the STROBE guidelines.

### 2.2. Participants 

We recruited people among in- and out-patients with chronic hemiparesis followed in the Physical Medicine and Rehabilitation Department of the Raymond Poincaré Hospital, Garches, France. The inclusion criteria were hemiparesis caused by a single stroke, age over 18 years, being able to perform several TUG tests alone and without the use of an assistive device, and being medically stable. The exclusion criteria were any other neurological, orthopedic, or medical disorders that might interfere with TUG test performance. 

### 2.3. Materials 

We used three-dimensional gait analysis, considered the gold standard for biomechanical gait assessment [6]. The optoelectronic motion capture system (Motion Analysis Corporation, Santa Rosa, CA, USA) was composed of eight infrared cameras (Eagle, 1.3 Mpixels) and 30 passive reflective markers placed on anatomical landmarks according to the Helen Hayes biomechanical full-body model, appropriate for gait analysis during straight-line walking and the TUG test (sampling frequency 100 Hz) [6,14,24,56]. The camera placement was optimized to capture movement during the TUG tests and the Straight-line walk. The same rigorous examiner positioned the markers for all the participants (11). 

### 2.4. Experimental Procedure 

All participants performed both locomotor tasks at a spontaneous, comfortable pace. The TUG test (3 trials) and the Straight-line walk (8 trials) were performed on a 10-meter walkway. The participants did not use any assistive devices or orthoses. The instrumented TUG was standardized; the participants wore the same trainers, the stool was adjusted to the distance between the head of the fibula and the floor, and the starting position was with the knees flexed at 100°, feet placed symmetrically, trunk straight, and arms by the sides. Marks and goniometric checks were used to ensure standardization across trials. Also, the participants were instructed to turn toward the paretic side for standardization purposes. The instruction was: “At the signal, stand up, walk at your own comfortable pace to the cone, turn around it to the right/left (depending on the side of the hemiparesis), and come back and sit down without using your arms”. These instructions were reiterated before each trial.

For the Straight-line walk, the participants were instructed to walk at a comfortable pace to the indicated mark: “At the signal, walk at your own comfortable pace to the mark”. 

We noted the following characteristics: sex, age, time since stroke, body mass index, and paretic side of hemiparesis. The same therapist performed the clinical assessments. Voluntary motor control was assessed with the Medical Research Council scale, spasticity with the Modified Ashworth Scale, balance with the Berg Balance Scale (BBS) from 0 to 56 [57,58], confidence in performing activities without losing balance with the Activities-specific Balance Confidence scale (ABC scale) from 0 (no confidence) to 100% (full confidence) [59,60], fear of falling with a visual analog scale (VAS) from 0 (not afraid) to 10 (extreme fear of falling), and the number of falls within the last 3 months. Figure 1 presents the experimental procedure.

### 2.5. Data Processing 

We used Cortex software (Cortex version 6.0, Motion Analysis Corporation, Santa Rosa, CA, USA) for data acquisition and processing. The first phase of data processing consisted of checking marker trajectories and performing conventional interpolation. A fourth-order Butterworth low-pass filter with a cut-off frequency of 6 Hz was applied. The second phase consisted of identifying gait cycle events, according to Perry’s method using Mokka software (Motion Kinematic and Kinetic Analyser, version 0.6.2) [61]. Foot strike and foot-off events were identified to divide the gait cycle into the swing and the support phases. For the TUG trials, the events of the TUG subtasks were added. The three navigational tasks have been presented in previous work [62,63]. The Go subtask (Go) began at the toe-off of the first step and ended with the first foot strike in the direction of the turn, the Turn subtask ended at the first foot strike lined up with the stool, and the Return subtask ended with foot strike of the last step before the turn to sit down. Figure 2 illustrates the data processing with markers tracked, gait events, and TUG events.

### 2.6. Parameters Analyzed

We analyzed STPs and joint kinematics during the Straight-line walk and TUG navigation subtasks. These included gait speed, step length, step width, %SSP and %DSP, peak hip and knee flexion and extension, and peak ankle plantar flexion and ankle dorsiflexion during the swing phase on both the paretic and non-paretic sides. 

We calculated COM displacements (using Dempster’s anthropometric table [64]) during the TUG navigational subtasks. COM displacement and velocity in the vertical and medio-lateral planes (Vert-COM, Vert-V, ML-COM, and ML-V, respectively) were used to indicate stability during each TUG navigational subtask (Go, Turn, and Return) [65,66]. We tracked the COM during the TUG navigational subtasks to analyze the gait trajectory deviation [24]. We calculated the trajectory deviation using Dynamic Time Warping (DTW) from a reference trajectory evaluated in healthy subjects [67,68]. DTW corresponds to the cumulative distance that minimizes the path traveled between two sets of trajectories. The Euclidean distance between each point of the two trajectories is first measured, and then an optimal matching of these distances is selected. The DTW result is unitless; the larger the value, the larger the deviation of the considered trajectory from the reference trajectory.

We also analyzed the time taken to perform the whole TUG and the time taken to complete the Go, Turn, and Return subtasks.

### 2.7. Statistical Analysis

We used the Jasp software (JASP 2021, Version 0.16, University of Amsterdam) for the statistical analysis. All significance levels were set at *p* < 0.05. We tested the normality of the variable distribution using the Shapiro–Wilk test. A post hoc analysis of power was performed with Gpower (Gpower version 3, University of Dusseldorf).

To compare kinematics between the TUG subtasks (Go, Turn, and Return) and the Straight-line walk, we performed a repeated measures ANOVA (η^2^ for effect size) as all kinematic variables followed a normal distribution. We applied a post hoc Holm test to the significant results (pholm). 

We analyzed correlations between STPs during each TUG subtask and the BBS score, ABC scale score, fear of falling VAS score, and number of falls using Pearson or Spearman tests, depending on the normality of the data. We compared STPs during the TUG navigational tasks and the Straight-line walk between fallers and non-fallers using an independent *t*-test (Cohen’s d was calculated to determine the effect size). 

We used multiple linear regression models with a stepwise ascending method to identify (i) the biomechanical parameters and (ii) the TUG subtask that best explained the TUG performance time variance. We first used a Pearson or Spearman test, depending on the normality of the data distribution, to select the input variables. We then entered variables that correlated significantly with TUG performance time into the regression models with a data transformation in case of non-normality (logarithmic 1 + x and Box-Cox) and the removal of collinear variables. The adjusted R^2^ regression coefficients were used to interpret the percentage variance of the TUG performance time, according to Domholdt [69]. 

## 3. Results

Twenty-nine individuals with chronic hemiparesis (mean age of 54.2 ± 12.2 years) participated in this study. Their characteristics are presented in Table 1. 

### 3.1. Comparisons between Tasks

The results of the comparisons between the TUG subtasks (Go, Turn, and Return) and the Straight-line walk are presented in Table 2. 

The values of all kinematic variables differed significantly between the subtasks and the Straight-line walk (*p* < 0.001), except for cadence, knee extension on the non-paretic side, and ankle plantar flexion on the paretic and non-paretic sides (*p* > 0.05). 

Gait speed was significantly slower, step length on the paretic side significantly shorter, and %DSP significantly longer during the Turn than during the Go and Return tasks (with no significant difference between Go and Return). All these variables differed significantly between the TUG subtasks and the Straight-line walk. Step width was significantly larger during the Turn than the Go, the Return, and the Straight-line walk. The %SSP on the paretic side and the %SSP on the non-paretic side were significantly shorter during the Turn than Go and Return. The %SSP on the paretic side was significantly shorter during the Go than during the Straight-line walk but not significantly different from the Return. 

Peak hip extension on the paretic and non-paretic sides differed significantly between the tasks, with progressively increasing values for Turn, Go, Return, and Straight-line walk (*p* < 0.001). Peak hip flexion on the paretic side was significantly greater during the Go, compared to the Return, the Turn, and the Straight-line walk (*p* < 0.001). Knee flexion on the paretic side was significantly lower during the Turn than during the Go, the Return, and the Straight-line walk (*p* < 0.001). Finally, the peak ankle dorsiflexion on the paretic side was also significantly lower during the Turn than during the Go and the Return (*p* < 0.001). Ankle plantar flexion did not differ between any of the TUG subtasks or the Straight-line walk (0.436). Figure 3 illustrates the results of the between-task comparisons.

### 3.2. Correlations between STPs and Clinical Variables and Differences between Fallers and Non-Fallers

Table 3 presents the correlations between spatio-temporal gait parameters during the three TUG subtasks, the BBS score, ABC scale score, fear of falling VAS score, and the number of falls. A larger number of STPs were significantly correlated with the number of falls and the BBS score than with the fear of falling VAS score and the ABC scale score. Gait speed during the Go and Return TUG subtasks was significantly correlated with all clinical variables. The %SSP during the Turn was also significantly correlated with most balance- and fall-related scale scores. Step width was not correlated with the balance-related or fear-related scale scores. 

Table 4 presents the differences of STP between fallers and non-fallers.

Fallers had a significantly higher gait speed during Turn and Return, longer paretic step length during Return, larger %SSP on the paretic side during Turn, and smaller %DSP during Turn and Return than non-fallers. Note that the effect sizes are very large.

### 3.3. Determinants of TUG Performance

The mean ± SD total TUG performance time was 19.3 ± 4.2 s. Table 5 presents the TUG performance time for each navigational subtask, the COM displacements (stability), and the DTW (locomotor trajectory deviation).

The TUG performance determinants found by the stepwise multiple linear regressions are presented in Table 6 and Table 7.

#### 3.3.1. Biomechanical Determinants 

The multiple regression model included the following TUG performance and biomechanical input variables (significantly correlated): Go Vert-V and Go % SSP on the paretic and non-paretic sides; Turn ML-V, Turn Number of Steps, Turn DTW, and Turn % SSP on the paretic and non-paretic sides; and Return Vert-V and Return % SSP on the paretic side. Gait speed and step length on paretic and non-paretic sides were collinear with the TUG performance and were thus excluded.

DTW during the Turn, % SSP on the paretic side during the Go, and Vert-V during the Go explained 71% of the Total TUG performance time variance (Table 5). The adjusted R^2^ value of 0.71 reflects the proportion of variance in TUG performance time accounted for by these variables (F(3.25) = 23.42; *p* < 0.001; Shapiro–Wilk p of residuals = 0.19).

#### 3.3.2. Determinant Subtasks

The multiple regression model of the TUG performance included the performance time for each navigational subtask (Go, Turn, and Return) as input variables (significantly correlated). 

The Turn performance time and the Go performance time together explained 82% of the variance of the total TUG performance time (F(2.26) = 65.88; *p* < 0.001; adjusted R^2^ = 0.82, Shapiro–Wilk p of residuals = 0.99) (Table 6).

## 4. Discussion

This study showed that people with hemiparesis adopt a cautious gait during complex locomotor tasks. Gait speed was lower, step length and %SSP on the paretic side were shorter, step width was larger, and %DSP was longer during the more complex navigation tasks (TUG navigational subtasks) than during the Straight-line walk. Several STPs during the TUG subtasks were correlated with balance disorders and fear of falling or balance confidence. STPs measured during the TUG distinguished fallers from non-fallers. TUG performance was determined by stability-related variables. 

### 4.1. Gait Adaptations according to the Nature of the Task

The differences in the STP values between the Turn and the Go and Return and the Straight-line walk suggest that the cautiousness of the gait increased with the complexity of the locomotor task (turning more complex than oriented gait and oriented gait more complex than straight-line walking). Studies in people with cerebellar ataxia or Parkinson’s disease found similar changes in STP values during turning and circular walking, and the authors concluded that these were adaptations to reduce instability [36,37]. Although adaptations of gait speed and step length during a complex walk on a treadmill have been observed in people with hemiparesis compared to non-disabled people [38], our study provides new insights into walking tasks frequently encountered in everyday life. The results suggest people with hemiparesis adopt a cautious gait during turns (or before or after a turn) in comparison with a simpler, straight-line walking task. 

These gait adaptations have been related to balance disorders and a fear of falling [26,41,42,43]. We found that most STP during the TUG navigational subtasks were significantly related to balance disorders and the occurrence of a recent fall than to fear of falling or balance confidence (ABC scale). The participants adapted their gait speed during oriented gait (to a target, i.e., the cone or the seat) according to their balance impairment, history of previous falls, and their fear and balance confidence. The paretic side %SSP (reflecting stability) was also shorter during the Turn and was correlated with balance impairment, previous falls, and the fear of falling, suggesting a cautious behavior. A study in people with multiple sclerosis and balance disorders also found that the turn velocity was strongly associated with balance confidence and gait abilities [70]. Therefore, balance disorders can potentiate the modulation of biomechanical parameters to increase gait stability. 

It is interesting that cadence did not change across the gait tasks. This can be explained by the previous finding that, above a gait speed of 0.33 m/s and a cadence of 90 steps/min (similar to the participants in the present study), people after stroke modulate their step length rather than their cadence to increase gait speed [71]. Hip extension, knee flexion, and ankle dorsiflexion peaks during the swing phase decreased concomitantly with the decrease in gait speed and step length during the Turn compared to the oriented gait Go and Return. Peak hip extension and peak knee flexion have already been identified as adjustable parameters related to gait speed in people with hemiparesis [72,73,74,75].

Finally, these results confirm the hypothesis that people with hemiparesis adapt their biomechanical parameters to the nature and complexity of the locomotor task to ensure their security. These findings have implications for rehabilitation, mainly that balance-related variables during gait tasks should be considered in rehabilitation planning. Assessing STPs during navigational tasks (especially turning) helps to understand gait strategies and the potential cautious gait behavior of people with hemiparesis. 

### 4.2. Gait Adaptations Differ between Fallers and Non-Fallers

The comparison of STPs during the TUG navigational subtasks and the Straight-line walk between fallers and non-fallers highlighted large differences during the Turn and Return, whereas STPs during the Straight-line walk did not differ between these groups. The higher gait speed, step length, and %SSP and lower %DSP in the participants who had previously sustained a fall during the Turn and Return suggest a less cautious behavior than that of those who had never fallen. 

Our results for the Straight-line walk differ from those of two previous studies that compared STPs during a straight-line walk between fallers and non-fallers in people with hemiparesis [23,46]. They showed that fallers had a significantly slower gait speed, lower cadence, shorter step length, and %SSP and a higher %DSP than people with hemiparesis who had never fallen. Note that the mean gait speed of the participants was 0.41 to 0.66 m/s, whereas the mean gait speed in our study was 0.81 ± 0.02 for fallers and 0.72 ± 0.02 for non-fallers. However, our results for the Straight-line walk are in accordance with those of Gimunova et al., (2022) in 210 community-dwelling older adults and Bourgarel et al. (2023) in 67 hospitalized older adults. Both studies found no differences in STPs between fallers and non-fallers (gait speed of around 0.8 m/s for Gimunova et al. and 0.5 m/s for Bourgarel et al.) [76,77]. 

Together, these results suggest that the locomotor behavior of people with hemiparesis with a mean gait speed of 0.7–0.8 m/s depends on the faller status for some locomotor tasks, especially complex tasks. These findings complete those of previous studies by our team that showed that fallers with chronic hemiparesis have a higher vertical velocity of their COM during turns and that their locomotor trajectory deviates more during the preparation phase of a turn than that of non-fallers [14,24].

### 4.3. Cautious Gait Associated with the Complexity of the Task

The Turn was the subtask of the TUG test with the largest differences in spatio-temporal parameter values compared to straight-line walking. Moreover, the regression analysis showed that the Turn performance time was the variable that most determined the total TUG performance time (63% of variance explained). The Go performance time was the next largest determinant, and the Return performance time was not a determinant. It could be suggested that people with hemiparesis perform the complex task of turning carefully, at the expense of timed performance. From a stability point of view, turning is a complex locomotor task that requires the adaptation of gait, particularly after a stroke (more steps and a longer time) [14,48]. This is supported by the fact that the turning capacity and stability (assessed by % SSP) are correlated [48]. 

The total TUG performance time was also essentially determined by biomechanical parameters during the Turn and during the previous task, Go (but not Return). The results show that the three parameters that best explained the TUG performance time (71% of the variance explained) were the locomotor trajectory during the Turn, followed by %SSP on the paretic side during the Go, and then vertical COM speed during the Go. This finding is in line with our previous studies, highlighting that people with hemiparesis are unstable during a turn and the oriented gait preceding it and that they deviate their locomotor trajectory to compensate for impaired balance [14,24]. 

Altogether, our results suggest that turning is a complex locomotor task for people with hemiparesis, requiring cautious gait behavior involving a stability management strategy at the expense of timed performance. These new findings should be considered in rehabilitation, which is almost exclusively focused on improving performance. People in the chronic phase of stroke may have a slow, safe gait or may walk at a faster speed with a risk of falls. 

Other studies found that people after a stroke modulate their walking speed to ensure safety during the execution of a complex task. For example, they may reduce gait speed as a precautionary strategy during obstacle-crossing to minimize the tripping risk [78]. The environment influences the gait speed of both people with and without hemiparesis; in complex environments, people in both groups reduce their gait speed and deviate their locomotor trajectory to anticipate obstacles and reduce the fall risk [35,79,80,81,82]. It could be suggested that, during the TUG, the perception of a visual target explicitly associated with a turning task impacts the gait of individuals with balance disorders who then adopt a cautious, anticipatory and reactive gait. The gait is modulated according to reactive and proactive processes to minimize the risks associated with instability (cautious gait) [66]. Our results suggest that people with hemiparesis are able to adapt gait parameters during and before complex navigational tasks. In line with this, studies have shown that previous experiences of instability enable proactive adjustments to be made to ensure stability [83,84]. We found that the gait parameters of the Go task (oriented gait preceding the Turn) were more decisive of the overall TUG performance time than the Return and that paretic side %SSP was significantly shorter during the Go (but not the Return) compared to the Straight-line walk. These findings provide evidence of cautious gait behavior during and in anticipation of the Turn. 

Finally, the biomechanical analysis of TUG test performance is relevant for understanding gait strategies. This approach provides essential information to guide rehabilitation by considering cautious behavior as well as timed performance. Future studies could use wearable devices to evaluate real-world complex navigation tasks. 

### 4.4. Strengths and Limitations 

The post hoc statistical analysis found a power between 0.96 and 0.99 (based on gait speed data), which allowed us to be confident that the sample size of 29 participants was sufficient to meet the aims of this study.

We quantified the biomechanical parameters using three-dimensional gait analysis. This instrument is very reliable in people with hemiparesis (intra-session and inter-session correlation coefficients between 0.85 and 0.99 for STPs and sagittal plane joint kinematics) [5]. Furthermore, to reduce errors, a single examiner followed the same standardized procedure and carried out all the experiments.

This study focused on joint kinematics in the sagittal plane as these parameters are associated with gait disorders in people with hemiparesis and are considered highly reliable [5,6,85]. However, the analysis of joint kinematics in other planes, or the addition of pelvic kinematic parameters, could increase the understanding of gait behavior in people with hemiparesis during different navigational tasks. Kinetic data could not be analyzed during the TUG in the laboratory. Additionally, the controlled laboratory environment may not fully capture the variability of locomotor behavior observed in real-world settings. Future research could explore the biomechanical adaptations of navigational tasks in a more diverse sample and in a real-world environment to enhance the generalizability of the findings.

We analyzed changes in the mean STP values, which reflect locomotor adaptations to ensure security, and we did not analyze STP variability, which reflects instability [41,86,87], because of the small number of gait cycles available for each subtask. The measurement of STP variability during complex locomotor tasks should be considered in future studies.

## 5. Conclusions

Although gait adaptations are well described after stroke, this study is the first to highlight changes in STPs depending on locomotor task complexity. The results confirm our hypothesis of cautious gait during complex tasks, in relation with the stability of people with chronic hemiparesis. Furthermore, non-fallers adopted a more cautious gait during complex locomotor tasks than fallers (no difference during a straight-line walk).

The biomechanical analysis of the TUG provides essential information to be considered alongside performance. The assessment of locomotor performance in people with hemiparesis should include dynamic stability. Rehabilitation should take into account this compromise between performance and safety and not systematically target improved performance, but promote a safe behavior during complex navigational tasks in fallers.

## Figures and Tables

**Figure 1 sensors-24-04241-f001:**
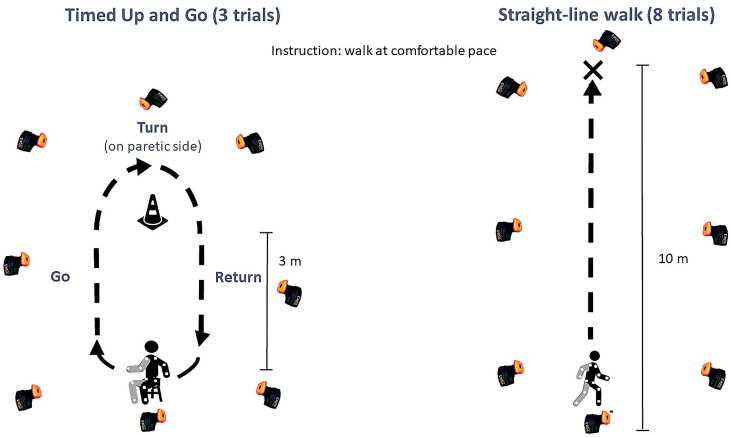
Experimental procedure.

**Figure 2 sensors-24-04241-f002:**
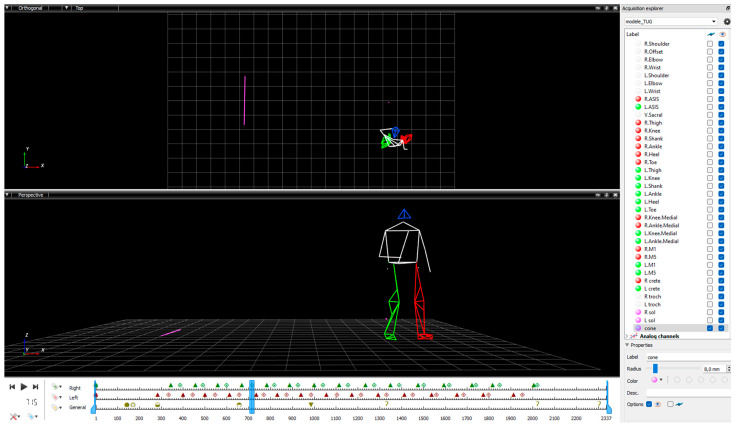
Illustration of the data processing with the Mokka software.

**Figure 3 sensors-24-04241-f003:**
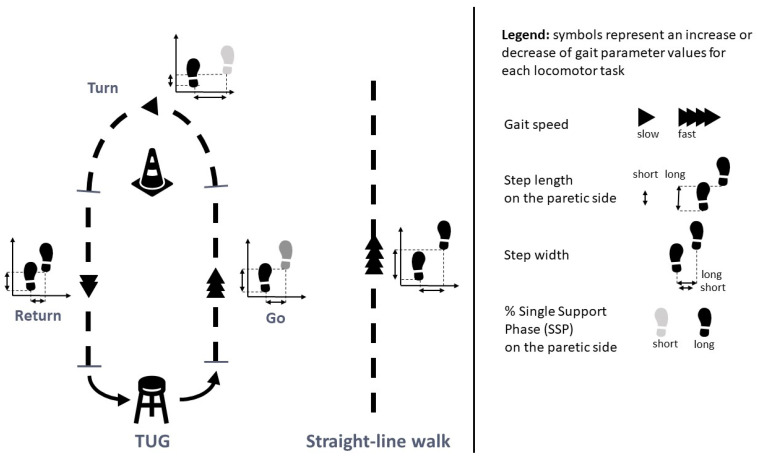
Illustration of the comparison of spatio-temporal gait parameters between the TUG subtasks (Go, Turn, and Return) and the Straight-line walk in people with hemiparesis.

**Table 1 sensors-24-04241-t001:** Participant characteristics (means ± standard deviations).

Sex (M/F)	Age (years)	Time Since Stroke (years)	BMI (kg/m^2^)	Hemiparetic Side (Right/Left)	BBS (0–56)	ABC Scale (0–100%)	Fear of Falling VAS (0–10)	Number of Falls within the Last 3 Months
18 M/11 F	54.2 ± 12.2	8.3 ± 6.1	25.6 ± 4.3	12 R/17 L	50.5 ± 2.3	76.3 ± 12.9	3.1 ± 3.2	0.65 ± 0.67

M/F: Male/Female. BMI: Body Mass Index. R/L: Right/Left. BBS: Berg Balance Scale. ABC scale: Activities-specific Balance Confidence Scale.

**Table 2 sensors-24-04241-t002:** Spatio-temporal parameters and joint kinematics during the TUG subtasks and the Straight-line walk (means ± standard deviations).

	TUG Subtasks			Straight-Line Walk	*p*-Value (η^2^)
Kinematic Criteria	Go	Turn	Return		
Gait speed (m/s)	0.41 ± 0.08	0.29 ± 0.6 a,d	0.40 ± 0.07	0.77 ± 0.02 c,e,f	<0.001 * (0.91)
Cadence (steps/min)	93.48 ± 11.13	92.60 ± 11.82	92.99 ± 10.49	91.23 ± 11.75	0.155 (0.06)
Step length P (m)	0.45 ± 0.08	0.28 ± 0.09 a,d	0.44 ± 0.07	0.51 ± 0.09 c,e,f	<0.001 * (0.68)
Step length NP (m)	0.42 ± 0.09	0.32 ± 0.09 a,d	0.43 ± 0.09	0.45 ± 0.01 e	<0.001 * (0.34)
Step width (m)	0.17 ± 0.05	0.22 ± 0.05 a,d	0.16 ± 0.05	0.19 ± 0.04 c,e,f	<0.001 * (0.63)
% SSP P (%)	28.45 ± 3.98	26.80 ± 4.30 a,d	29.20 ± 3.69	30.04 ± 4.05 c,e	<0.001 * (0.45)
% SSP NP (%)	39.90 ± 3.36	36.52 ± 4.35 a,d	39.15 ± 2.83	40.76 e,f ± 3.71	<0.001 * (0.42)
% DSP (%)	31.83 ± 5.76	36.69 ± 7.20 a,d	31.84 ± 5.00	29.20 ± 5.15 c,e,f	<0.001 * (0.59)
Hip flexion P (°)	40.57 ± 10.59	35.93 ± 9.64 a	36.42 ± 9.60 b	35.75 ± 9.63 c	<0.001 * (0.41)
Hip extension P (°)	−2.83 ± 8.54	−5.47 ± 9.34 a,d	−1.15 ± 8.32 b	−0.13 ± 8.60 c,e	<0.001 * (0.59)
Knee flexion P (°)	44.13 ± 8.58	40.15 ± 8.43 a,d	44.28 ± 10.36	44.25 ± 12.12 e	<0.001 * (0.26)
Knee extension P (°)	−2.01 ± 7.07	−2.62 ± 7.46	−1.14 ± 6.27	−0.90 ± 5.96 e	0.024 * (0.11)
Ankle dorsiflexion P (°)	16.86 ± 6.62	14.86 ± 6.28 a,d	16.40 ± 6.76	15.97 ± 7.22	<0.001 * (0.20)
Ankle dorsiflexion swing phase P (°)	1.26 ± 7.28	0.18 ± 8.67	0.63 ± 7.39	1.75 ± 5.98 e	0.033 * (0.10)
Ankle plantar flexion P (°)	10.37 ± 7.79	9.85 ± 9.54	10.82 ± 8.60	10.24 ± 7.30	0.436 (0.03)
Hip flexion NP (°)	47.10 ± 8.42	43.80 ± 8.25 a	44.80 ± 8.13 b	45.50 ± 8.10 c,e	<0.001 * (0.34)
Hip extension NP (°)	4.50 ± 8.77	3.10 ± 8.61 a,d	5.60 ± 8.66 b	6.30 ± 8.54 c,e	<0.001 * (0.49)
Knee flexion NP (°)	70.50 ± 5.03	69.40 ± 5.61 a	69.90 ± 5.14	69.00 ± 4.64 c,f	<0.001 * (0.19)
Knee extension NP (°)	−5.70 ± 5.23	5.10 ± 5.14	−5.10 ± 5.56	−6.00 ± 5.95	0.091 (0.07)
Ankle dorsiflexion NP (°)	23.70 ± 3.71	20.00 ± 3.58 a,d	22.00 ± 3.49 b	21.40 ± 3.90 c,e	<0.001 * (0.51)
Ankle dorsiflexion swing phase NP (°)	16.30 ± 6.17	13.80 ± 6.09	15.00 ± 6.89	7.60 ± 3.26 c,e,f	<0.001 * (0.45)
Ankle plantar flexion NP (°)	11.30 ± 6.16	9.90 ± 5.61	10.70 ± 6.21	9.70 ± 5.63	0.079 (0.08)

* Overall significant difference (*p* < 0.05). a, significant difference between Go and Turn (*p* < 0.05). b, significant difference between Go and Return (*p* < 0.05). c, significant difference between Go and Straight-line walk (*p* < 0.05). d, significant difference between Turn and Return (*p* < 0.05). e, significant difference between Turn and Straight-line walk (*p* < 0.05). f, significant difference between Return and Straight-line walk (*p* < 0.05). η^2^: Effect size. All joint kinematic data (hip, knee, and ankle) refer to peak values. For hip and knee extension, negative signs indicate a peak range of motion deficit. P: Paretic side. NP: Non-paretic side. % SSP: Percentage duration of single support phase (percentage of the gait cycle). % DSP: Percentage duration of double support phase (percentage of the gait cycle).

**Table 3 sensors-24-04241-t003:** Correlations between the spatio-temporal parameters of the TUG subtasks and clinical variables: correlation coefficient (*p*-value).

	Berg Balance Scale	Fear of Falling VAS	ABC Scale	Number of Falls
Gait speed Go (m/s)	0.41 (0.025)	−0.40 (0.033)	0.42 (0.022)	0.38 (0.044)
Step length P Go (m)	0.45 (0.013)	−0.30 (0.113)	0.35 (0.064)	0.27 (0.149)
Step width Go (m)	−0.35 (0.060)	0.37 (0.051)	−0.08 (0.696)	0.05 (0.810)
% SSP P Go (%)	0.38 (0.042)	−0.27 (0.157)	0.21 (0.278)	0.33 (0.083)
% DSP Go (%)	−0.34 (0.073)	0.22 (0.253)	−0.26 (0.180)	−0.43 (0.021)
Gait speed Turn (s)	0.34 (0.071)	−0.34 (0.072)	0.18 (0.340)	0.49 (0.007)
Step length P Turn (m)	0.45 (0.014)	−0.26 (0.172)	0.26 (0.164)	0.25 (0.195)
Step width Turn (m)	−0.35 (0.059)	0.15 (0.421)	0.01 (0.947)	−0.08 (0.695)
% SSP P Turn (%)	0.38 (0.045)	−0.40 (0.033)	−0.24 (0.215)	0.38 (0.042)
% DSP Turn (%)	−0.31 (0.099)	0.28 (0.138)	−0.26 (0.169)	−0.46 (0.011)
Gait speed Return (s)	0.39 (0.037)	−0.45 (0.013)	0.42 (0.023)	0.44 (0.016)
Step length P Return (m)	0.35 (0.063)	−0.26 (0.180)	0.35 (0.065)	0.42 (0.023)
Step width Return (m)	−0.29 (0.118)	0.32 (0.091)	−0.02 (0.917)	−0.01 (0.969)
% SSP P Return (%)	0.33 (0.075)	−0.34 (0.071)	0.27 (0.160)	0.36 (0.056)
% DSP Return (%)	−0.24 (0.202)	0.22 (0.247)	−0.28 (0.141)	−0.46 (0.011)

P, paretic. % SSP, Percentage single support phase. % DSP, Percentage double support phase. VAS, visual analog scale. ABC, Activities-specific Balance Confidence scale.

**Table 4 sensors-24-04241-t004:** Comparison of spatio-temporal gait parameters during TUG subtasks and the Straight-line walk between fallers and non-fallers.

	Fallers	Non-Fallers	*p*-Value (Effect Size)
Gait speed Go (m/s)	0.44 (0.09)	0.38 (0.07)	0.063 (0.72)
Gait speed Turn (m/s)	0.31 (0.07)	0.25 (0.04)	0.012 (1.0) *
Gait speed Return (m/s)	0.43 (0.08)	0.37 (0.05)	0.013 (0.99) *
Gait speed Straight-line (m/s)	0.81 (0.02)	0.72 (0.02)	0.202 (0.49)
Step length P Go (m)	0.47 (0.09)	0.43 (0.07)	0.208 (0.48)
Step length P Turn (m)	0.29 (0.11)	0.25 (0.07)	0.264 (0.43)
Step length P Return (m)	0.47 (0.07)	0.40 (0.05)	0.017 (0.95) *
Step length P Straight-line (m)	0.51 (0.12)	0.52 (0.05)	0.840 (−0.08)
Cadence Go (steps/min)	95.1 (11.9)	91.5 (10.2)	0.401 (0.32)
Cadence Turn (steps/min)	94.5 (11.8)	90.3 (11.9)	0.348 (0.36)
Cadence Return (steps/min)	94.5 (11.2)	91.1 (9.6)	0.396 (0.32)
Cadence Straigth-line (steps/min)	92.1 (14.2)	90.2 (8.3)	0.679 (0.16)
Step width Go (m)	0.18 (0.05)	0.16 (0.05)	0.583 (0.21)
Step width Turn (m)	0.23 (0.05)	0.22 (0.04)	0.666 (0.16)
Step width Return (m)	0.16 (0.05)	0.15 (0.04)	0.605 (0.19)
Step width Straigth-line (m)	0.20 (0.06)	0.19 (0.02)	0.508 (0.25)
% SSP P Go (%)	29.4 (4.3)	27.2 (3.23)	0.135 (0.57)
% SSP P Turn (%)	28.3 (4.5)	24.9 (28.3)	0.046 (0.78) *
% SSP P Return (%)	30.3 (3.9)	27.8 (2.9)	0.072 (0.69)
% SSP P Straigth-line (%)	30.7 (4.3)	29.2 (3.7)	0.323 (0.38)
% DSP Go (%)	30.3 (5.8)	34.6 (5.3)	0.052 (−0.76)
% DSP Turn (%)	33.9 (6.2)	40.1 (6.2)	0.013 (−0.99) *
% DSP Return (%)	30.3 (5.2)	34.5 (3.8)	0.023 (−0.90) *
% DSP Straight-line (%)	27.7 (4.2)	31.1 (5.7)	0.075 (−0.69)

Data are mean (SD) except for p-value (effect size). * significant difference between fallers and non-fallers (*p* < 0.05). Effect size: Cohen’s d. P: paretic side. % SSP: Percentage duration of single support phase (percentage of the gait cycle). % DSP: Percentage duration of Double support phase (percentage of the gait cycle).

**Table 5 sensors-24-04241-t005:** TUG performance, stability, and trajectory deviation for each TUG subtask (means ± standard deviations).

	Navigational Subtasks of the TUG		
TUG Performance, Stability, and Trajectory Criteria	Go	Turn	Return
TUG performance (s)	4.6 ± 1.0	3.2 ± 0.8	3.8 ± 0.9
Vert-COM (cm)	4.5 ± 1.1	3.6 ± 0.8	4.6 ± 0.9
Vert-V (cm/s)	18.9 ± 4.7	15.5 ± 3.6	20.5 ± 4.3
ML-COM (cm)	8.9 ± 1.8	19.0 ± 4.4	9.2 ± 2.0
ML-V (cm/s)	23.0 ± 4.4	30.1 ± 5.9	22.6 ± 4.6
DTW	5894.5 ± 4699.8	7074.1 ± 4301.6	7131.6 ± 6970.3

TUG: Timed Up and Go. Vert-COM: Range of motion of the center of mass in the vertical plane. Vert-V: Velocity of motion of the center of mass in the vertical plane. ML-COM: Range of motion of the center of mass in the medio-lateral plane. ML-V: Velocity of motion of the center of mass in the medio-lateral plane. DTW: Dynamic time warping.

**Table 6 sensors-24-04241-t006:** Multiple linear regression model of the total TUG performance time and biomechanical parameters of the TUG subtasks.

Model	Adjusted R^2^	Standard Error	*p*
DTW Turn	0.54	8.33	<0.001
DTW Turn + %SSP P Go	0.64	10.92	<0.001
DTW Turn + %SSP P Go + Vert-V Go	0.71	10.16	<0.001

DTW Turn: Dynamic time warping during the Turn. %SSP P Go: Percentage duration of single support phase on the paretic side during the Go (percentage of the gait cycle). Vert-V Go: Vertical center of mass displacement velocity.

**Table 7 sensors-24-04241-t007:** Multiple linear regression model of the total TUG performance time and the Go, Turn, and Return performance times.

Model	Adjusted R^2^	Standard Error	*p*
Turn performance time	0.63	1.90	<0.001
Turn performance + Go performance time	0.82	1.66	<0.001

## Data Availability

Data will be made available upon reasonable request to the corresponding author.

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
