# Peer review of "Cautious Gait during Navigational Tasks in People with Hemiparesis: An Observational Study"

_sensors, 2024, doi:10.3390/s24134241_

Round 1

Reviewer 1 Report

Comments and Suggestions for Authors

The authors used three-dimensional gait analysis to measure spatiotemporal gait metrics in patients with hemiparesis after stroke as they performed TUG and straight-line walk tests. The key message of the study is that participants adopted a more cautious (for example, slower walking speed and shorter step length) approach during the more complex TUG test compared to the straight-line walk test.  

The finding that gait metrics were “worse” or “more cautious” in hemiparetic patients during complex locomotion tasks may be interesting in and of itself but could be made more significant if related to clinical utility. The authors have hinted throughout the paper that the more cautious gait may be a consequence of reduced stability. To add clinical utility to this work, I believe it would be valuable to extrapolate this to falls risk. That is, hemiplegic patients when made to do complex locomotion tasks have “worse” (slower walking speed, shorter step length) gait metrics because they feel unsteady and may be more likely to fall. If the authors have any data on the falls data of the participants of their study, it may be insightful to compare the TUG (complex locomotion task) gait metrics of those participants who had falls compared to those who did not (or who had less falls). It may also be helpful to briefly compare the authors’ data with literature findings on the gait metrics of patients who have had falls.

A future implication of this may be in identifying participants who are at a high risk of falls by analysing their gait during complex locomotion tasks, and then targeting these patients with rehabilitation to reduce falls risk.

The study was reasonably well designed. Gold-standard gait analysis techniques were used. A sample size of 29 participants is adequate. The TUG and straight-line walk tests are well established. The results are interpreted appropriately and the discussion is well written.

The language and grammar are appropriate.

Author Response

Reviewer 1

The authors used three-dimensional gait analysis to measure spatiotemporal gait metrics in patients with hemiparesis after stroke as they performed TUG and straight-line walk tests. The key message of the study is that participants adopted a more cautious (for example, slower walking speed and shorter step length) approach during the more complex TUG test compared to the straight-line walk test.  

The finding that gait metrics were “worse” or “more cautious” in hemiparetic patients during complex locomotion tasks may be interesting in and of itself but could be made more significant if related to clinical utility. The authors have hinted throughout the paper that the more cautious gait may be a consequence of reduced stability. To add clinical utility to this work, I believe it would be valuable to extrapolate this to falls risk. That is, hemiplegic patients when made to do complex locomotion tasks have “worse” (slower walking speed, shorter step length) gait metrics because they feel unsteady and may be more likely to fall. If the authors have any data on the falls data of the participants of their study, it may be insightful to compare the TUG (complex locomotion task) gait metrics of those participants who had falls compared to those who did not (or who had less falls). It may also be helpful to briefly compare the authors’ data with literature findings on the gait metrics of patients who have had falls.

Thanks to the reviewer for his/her recommendations which helped us improve the quality of our article.

The secondary aim of this study was to explore the correlations between spatio-temporal gait parameters during the TUG navigational tasks and clinical measures of balance, confidence and falls. The results showed significant correlations mainly for gait speed and temporal parameters (% single and double support phase) during each sub-task. We suggested that these correlations and the differences in gait parameters depending on the complexity of the task argue in favor of cautious behavior.

Thank you for the suggestion to add comparisons of spatio-temporal gait parameters during the TUG navigational tasks between fallers and non-fallers. We had the data required to perform these comparisons. The results showed significant differences in gait speed during Turn and Return (higher for fallers), step length on paretic side during Return (longer for fallers), % single support phase on the paretic side during Turn (longer for fallers) and % double support phase during Turn and return (shorter for fallers). In contrast, none of the spatio-temporal gait parameters during straight-line differed between fallers and non-fallers. These complementary results are very interesting and we thank you again for the suggestion. Combined with other results, these new results suggest that hemiparetic non-faller individuals seem to adopt a more cautions gait than hemiparetic non-faller individuals, especially during complex walking tasks. We have added these results to the manuscript.

Please see Abstract p1, l21-22 : “ compare spatio-temporal gait parameters between fallers and non-fallers”

Please see p2, l81-83 “People with hemiparesis who fall have a slower gait, slower cadence, and larger SSP percentage asymmetry compared to non-fallers during straight-line walking [23, 46]. »

Please see p2, l93-94 : “ However, STP during turn and navigational tasks in fallers and non-fallers with hemiparesis are unknown. “

Please see p3, l111-112 : “and to compare STP during the TUG navigational tasks and Straight-line walk between fallers and non-fallers. “

Please see p6, l250-252 : “ We compared STP during the TUG navigational tasks and the Straight-line walk between fallers and non-fallers using an independent t-test (Cohen’s d was calculated to determine the effect size). “

Please see Table 4 p11

Please see p11-12, l338-341 “Faller individuals had a significantly higher gait speed during Turn and Return, longer paretic step length during Return, larger %SSP on the paretic side during Turn and smaller %DSP during Turn and Return than non-faller individuals. Note that the effect sizes are very large.”

Please see p14, l438-460, New paragraph “ Gait adaptations differ between fallers and non-fallers.

The comparison of STP during the TUG navigational subtasks and the Straight-line walk between fallers and non-fallers ….. “

These results have clinical implications: gait behavior during complex locomotion tasks may help to identify people at risk of falls, who could then undergo targeted rehabilitation programs to reduce falls risk.

We added a sentence in conclusion associating these results and perspectives.

Please see p16, l544-547 : “ Furthermore, STP during the TUG, but not the Straight-line walk, distinguished fallers from non-fallers. The results also showed that non-fallers adopted a more cautious gait during complex locomotor tasks than fallers. “ 

Please see p16, l553-555: “Finally, the assessment of locomotor performance in people with hemiparesis should include dynamic stability; rehabilitation could aim not only to improve performance but also teach safe gait behavior during complex navigational tasks, especially in fallers.“

The study was reasonably well designed. Gold-standard gait analysis techniques were used. A sample size of 29 participants is adequate. The TUG and straight-line walk tests are well established. The results are interpreted appropriately and the discussion is well written. The language and grammar are appropriate.

Thank you

Reviewer 2 Report

Comments and Suggestions for Authors

This study examines movement and balance problems in individuals with hemiparesis, focusing on the standardized testing Timed 14 Up and Go (TUG), which is related to oriented walking and obstacle avoidance. The main aim of the study is to compare spatio-temporal gait parameters, especially careful gait indicators, during different TUG tasks (walk, turn, return) and straight-line walking in people with hemiparesis. This study also aims to analyze the relationship between TUG performance and balance measures and to identify biomechanical factors that influence TUG performance. The obtained results indicate that subjects with hemiparesis have reduced gait speed, stride length, and percentage of single support phase during the turning part compared to walking, returning, and walking in a straight line. Conversely, it has been observed that the stride width and percentage of the double support phase increase during the turning part. part. The work states that turning performance and trajectory deviation, followed by the percentage of one support phase on the paretic side, as well as the vertical centre of mass velocity during walking are important factors determining the operation time of the TUG. The findings highlight the importance of understanding nuanced gait patterns and balance problems in individuals with hemiparesis, which may inform rehabilitation strategies and interventions to improve mobility and function. The work is interesting and provides scientific insights into stroke-related behavioural adaptations during complex locomotor tasks. On the other hand, the work confirms already known observations but does not create completely new knowledge.

but I have a few comments and questions:

1)      I recommend removing the sentence "During complex locomotor tasks elderly and people with Parkinson's disease adopt a cautious gait" from the abstract since the publication does not examine these dysfunctions and only the conclusions contain a few statements about the knowledge presented in those areas. It will be very confusing for the reader.

2) I suppose that it is important to announce that an instrumental analysis of TUG and straight-line test was performed using a motion capture system already in the abstract and even in the keywords. It does mention "biomechanical analysis," which could involve various methods for assessing movement patterns and kinetics, including motion capture systems, but it needs clarification.

3)      It is written that "30 passive reflective markers placed on anatomical landmarks according to the Helen Hayes model", it is not clear why such a set of markers was chosen. Is this a full-body marker set? Why was such an arrangement of markers necessary for TUG and straight–line walk analysis? Justify and explain.

4)      It is worth presenting the experimental procedure graphically, then it would be easier to understand the sequence of activities and the set-up of each step.

5)      The Data processing and parameter analysis paragraphs require a graphical or visual example of data processing to better understand how individual parameters and events were analyzed. It is also important to justify why one or another method of data processing and analysis was applied.

6)      Was the sample size relevant to the significance of the study results calculated? Explain and justify.

7)      Fig. 1 belongs to the Materials and Methods part.

8)      In my opinion, the discussion is about already known results confirmed by this study. So what is the contribution of the authors then? It is necessary to highlight novelty, relevance and practical value. Also, emphasize the challenges and pay attention to the most relevant results obtained.

9)      Part of the conclusion „Biomechanical analysis of TUG test performance is relevant for understanding gait strategies. This approach provides essential information to guide rehabilitation by considering cautious behaviour as well as timed performance. Future studies could use wearable devices to evaluate real-world complex navigation tasks“  belongs to the discussion part.

10)  Conclusions need to be very clearly linked to results, i.e. with specific significant values obtained confirming the insights.

11)  Editorial corrections must be made, text aligned, spaces fixed, etc.

In summary, given the emphasis on "Cautious gait" in the title of the article, this emphasis needs to be maintained consistently throughout the entire publication, which currently feels lacking. Nevertheless, the study demonstrates potential and contributes valuable insights to academic knowledge.

Author Response

This study examines movement and balance problems in individuals with hemiparesis, focusing on the standardized testing Timed Up and Go (TUG), which is related to oriented walking and obstacle avoidance. The main aim of the study is to compare spatio-temporal gait parameters, especially careful gait indicators, during different TUG tasks (walk, turn, return) and straight-line walking in people with hemiparesis. This study also aims to analyze the relationship between TUG performance and balance measures and to identify biomechanical factors that influence TUG performance. The obtained results indicate that subjects with hemiparesis have reduced gait speed, stride length, and percentage of single support phase during the turning part compared to walking, returning, and walking in a straight line. Conversely, it has been observed that the stride width and percentage of the double support phase increase during the turning part. part. The work states that turning performance and trajectory deviation, followed by the percentage of one support phase on the paretic side, as well as the vertical centre of mass velocity during walking are important factors determining the operation time of the TUG. The findings highlight the importance of understanding nuanced gait patterns and balance problems in individuals with hemiparesis, which may inform rehabilitation strategies and interventions to improve mobility and function. The work is interesting and provides scientific insights into stroke-related behavioural adaptations during complex locomotor tasks. On the other hand, the work confirms already known observations but does not create completely new knowledge.

but I have a few comments and questions

Thanks to the reviewer for his/her recommendations which helped us improve the quality of our article.

1)      I recommend removing the sentence "During complex locomotor tasks elderly and people with Parkinson's disease adopt a cautious gait" from the abstract since the publication does not examine these dysfunctions and only the conclusions contain a few statements about the knowledge presented in those areas. It will be very confusing for the reader.

As suggested by the reviewer, we removed this sentence.

Please see p1, l17-18 : “We hypothesized that subjects with hemiparesis adopt a cautious gait during complex locomotor tasks.”

2) I suppose that it is important to announce that an instrumental analysis of TUG and straight-line test was performed using a motion capture system already in the abstract and even in the keywords. It does mention "biomechanical analysis," which could involve various methods for assessing movement patterns and kinetics, including motion capture systems, but it needs clarification.

Thank you for pointing this out, we have now clarified the abstract.  

Please see p1, l23 : “Hemiparetic patients performed TUG and Straight-line walk analyzed using a motion capture system”

Please see p1, l32-33 : « Keywords: Cautious gait, navigation, stroke, stationarity, Timed Up and Go, motion capture, kinematics »

3)      It is written that "30 passive reflective markers placed on anatomical landmarks according to the Helen Hayes model", it is not clear why such a set of markers was chosen. Is this a full-body marker set? Why was such an arrangement of markers necessary for TUG and straight–line walk analysis? Justify and explain.

The Helen Hayes model is a biomechanical full-body marker set of 12 segments which was associated with the Cortex software of our motion capture system: Motion Analysis (Motion Analysis Corporation, Santa Rosa, CA, USA). The justification for the use of this model comes from the reference Kadaba 1990 (Kadaba, M.P., Ramakrishnan, H.K., Wootten, M.E., 1990. Measurement of lower extremity kinematics during level walking. J. Orthop. Res. 8, 383–92).

The following figure from the reference manual Orthotrack Gait Analysis Sofware presents the marker set:

This biomechanical model is appropriate and widely used for the analysis of spatio-temporal gait parameters during straight-line walking and the TUG (previous studies Bonnyaud 2015, 2016).

Bonnyaud C, Pradon D, Bensmail D, Roche N. Dynamic Stability and Risk of Tripping during the Timed Up and Go Test in Hemiparetic and Healthy Subjects. Baron J-C, éditeur. PLOS ONE. 2015;10:e0140317. doi: 10.1371/journal.pone.0140317

Bonnyaud C, Roche N, Van Hamme A, Bensmail D, Pradon D. Locomotor Trajectories of Stroke Patients during Oriented Gait and Turning. Baron J-C, éditeur. PLOS ONE. 2016;11:e0149757. doi: 10.1371/journal.pone.0149757

Specifications have been added to the manuscript as recommended.

Please see p4, l146-151 : “The optoelectronic motion capture system (Motion Analysis Corporation, Santa Rosa, CA, USA) was composed of eight infrared cameras (Eagle, 1.3 Mpixels) and 30 passive reflective markers placed on anatomical landmarks according to the Helen Hayes biomechanical full-body model, appropriate for gait analysis during straight-line walking and the TUG test (sampling frequency 100 Hz) [6,14,24,55].”

4)      It is worth presenting the experimental procedure graphically, then it would be easier to understand the sequence of activities and the set-up of each step.

Please see Figure 1 p4

5)      The Data processing and parameter analysis paragraphs require a graphical or visual example of data processing to better understand how individual parameters and events were analyzed. It is also important to justify why one or another method of data processing and analysis was applied.

As suggested, we added a figure to illustrate the data processing (markers followed, gait events and TUG events).

Please see Figure 2 p5

Please see p4, l203-204 : ” Figure 2 illustrates data processing with markers tracked, gait events and TUG events. ”

6)      Was the sample size relevant to the significance of the study results calculated? Explain and justify.

The results of this study showed significant differences with large effect sizes (comparisons between locomotor tasks, and also between fallers and non-fallers as asked by reviewer 1). As asked, we conducted a statistical post hoc analysis of gait speed during Go, Turn, Return and straight-line walking using Gpower software (Gpower version 3, University of Dusseldorf). The results showed a post-hoc statistical power between 0.96 and 0.99. We can thus be confident that the sample size was appropriate to meet our objectives. This point has been added in the manuscript.

Please see p6, l242-243 : ” A post-hoc analysis of power was performed with Gpower (Gpower version 3, University of Dusseldorf). ”

Please see p16, l513-515 : ” Post-hoc statistical analysis found a power between 0.96 and 0.99 (based on gait speed data); which allowed us to be confident that the sample size of 29 participants was sufficient to meet the aims of this study. ”

7)      Fig. 1 belongs to the Materials and Methods part.

This figure aimed to illustrate the results of the comparisons of the main spatio-temporal gait parameters between the different locomotor tasks (Go, Turn, Return, straight-line walk). The legend shows if parameter values were high or low depending on the number of triangles for gait speed, depending on arrow size for step length and step width, depending on color (grey/black) for single support phase.

The title and the figure have been changed to improve understanding for readers.

Please see p9, l297-299 : ”  Figure 3. Illustration of comparison of spatio-temporal gait parameters between the TUG subtasks (Go, Turn and Return) and the Straight-line walk in people with hemiparesis.” 

Please see Figure 3

8)      In my opinion, the discussion is about already known results confirmed by this study. So what is the contribution of the authors then? It is necessary to highlight novelty, relevance and practical value. Also, emphasize the challenges and pay attention to the most relevant results obtained.

We have rework the discussion taking into account your comment. The emphasis has been placed on the important elements, the clinical implications for rehabilitation have been presented, and a few unhelpful elements have been removed. A paragraph relating to the new analysis requested by reviewer 1 was also added. This provides new knowledge about distinguishing between fallers and non-fallers using spatio-temporal parameters measured during a complex locomotor task (especially turn).

Please see p13 to 15

Important sentences:

p13-14, l405-409: ”  Although adaptations of gait speed and step length during a complex walk on a treadmill have been observed in people with hemiparesis compared to non-disabled people [38], our study provides new insights into walking tasks frequently encountered in everyday life. The results suggest people with hemiparesis adopt a cautious gait during turns (or before or after a turn) in comparison with a simpler, straight-line walking task. ”  

p15, l480-485 ”  Altogether, our results suggest that turning is a complex locomotor task for people with hemiparesis, requiring cautious gait behavior involving a stability management strategy at the expense of timed performance. These new findings should be considered in rehabilitation, which is almost exclusively focused on improving performance. People in the chronic phase of stroke may have a slow, safe gait or may walk at a faster speed with a risk of falls. ” 

9)      Part of the conclusion, “Biomechanical analysis of TUG test performance is relevant for understanding gait strategies. This approach provides essential information to guide rehabilitation by considering cautious behaviour as well as timed performance. Future studies could use wearable devices to evaluate real-world complex navigation tasks“  belongs to the discussion part.

As suggested, this part has been placed in the discussion section.

Please see p15, l479-482.

10)  Conclusions need to be very clearly linked to results, i.e. with specific significant values obtained confirming the insights.

As suggested, we specified the variables statistically modified and the significant correlations to justify conclusion. The conclusion part has been considerably rewritten.

Please see p16, l538-555 ”  Although gait adaptations have been well described after stroke, this study is the first to highlight changes in STP depending on locomotor task complexity. The results showed a significant decrease in gait speed, step length and paretic %SSP and an increase in %DSP and step width during the TUG navigational subtasks (especially turning) compared to the Straight-line walk in people with hemiparesis. These adaptations were related to balance abilities and falls. These results enable us to conclude that individuals with stroke adopt a cautious gait during complex navigational tasks to ensure stability. Furthermore, STP during the TUG, but not the Straight-line walk, distinguished fallers from non-fallers. The results also showed that non-fallers adopted a more cautious gait during complex locomotor tasks than fallers.

This study also showed that the TUG test is not only an indicator of walking performance but, when combined with motion analysis, these complex navigational tasks can be used to identify biomechanical locomotor adaptations involving a compromise between stability, trajectory, kinematics and performance to ensure safety, especially in non-fallers.

Finally, the assessment of locomotor performance in people with hemiparesis should include dynamic stability; rehabilitation could aim not only to improve performance but also teach safe gait behavior during complex navigational tasks, especially in fallers. ” 

11)  Editorial corrections must be made, text aligned, spaces fixed, etc.

We corrected these errors. Thank you for your attention.

In summary, given the emphasis on "Cautious gait" in the title of the article, this emphasis needs to be maintained consistently throughout the entire publication, which currently feels lacking. Nevertheless, the study demonstrates potential and contributes valuable insights to academic knowledge.

We have taken into consideration your comments, we thank you for your advise to improve the article.

Round 2

Reviewer 1 Report

Comments and Suggestions for Authors

The authors have adequately responded to the suggestions in my prior review.

Author Response

Reviewer 1 :

The authors have adequately responded to the suggestions in my prior review.

Thank you for your recommendations 

Reviewer 2 Report

Comments and Suggestions for Authors

The authors have significantly improved their publication and I believe it will be interesting and valuable to readers.

I have one note about the conclusions. I think they are too long and general in nature. The entire discussion and summary are already in the Discussion section. And the conclusions must present clear statements that follow from the obtained results. It is the confirmation or refutation of a hypothesis that is linked very clearly to the implications of the research results. I suggest the authors re-examine the conclusions presented.

Good luck and successful future work.

Author Response

Reviewer 2 :

The authors have significantly improved their publication and I believe it will be interesting and valuable to readers.

I have one note about the conclusions. I think they are too long and general in nature. The entire discussion and summary are already in the Discussion section. And the conclusions must present clear statements that follow from the obtained results. It is the confirmation or refutation of a hypothesis that is linked very clearly to the implications of the research results. I suggest the authors re-examine the conclusions presented.

Good luck and successful future work.

Thank you for your recommendations.

As suggested, the conclusion has been shortened and rewritten in part to focus more on clear statements responding to our hypothesis and implications in rehabilitation.

Please see p16, l538-547:  “Although gait adaptations are well described after stroke, this study is the first to highlight changes in STP depending on locomotor task complexity. The results confirm our hypothesis of cautious gait during complex tasks, in relation with stability of people with chronic hemiparesis. Furthermore, non-fallers adopted a more cautious gait during complex locomotor tasks than fallers (no difference during straight-line walk). The biomechanical analysis of the TUG provides essential information to be consider alongside performance. The assessment of locomotor performance in people with hemiparesis should include dynamic stability. Rehabilitation should take into account this compromise between performance and safety and, not systematically target improved performance, but promote a safe behavior during complex navigational tasks in fallers.”